# Individual and community level factors associated with anemia among children 6—59 months of age in Ethiopia: A further analysis of 2016 Ethiopia demographic and health survey

**Menaseb Gebrehaweria Gebremeskel**[1]*, **Afework Mulugeta**[2], **Abate Bekele**[3], **Lire Lemma**[4], **Muzey Gebremichael**[1], **Haftay Gebremedhin**[1], **Berhe Etsay**[1], **Tesfay Tsegay**[5], **Yared Haileslasie**[5], **Yohannes Kinfe**[6], **Fre Gebremeskel**[1], **Letemichael Mezgebo**[1], **Selam Shushay**[7]

1 Department of Public Health, College of Medicine and Health Sciences, Adigrat University, Tigray, Ethiopia, 2 Department of Nutrition, School of Public Health, College of Health Sciences, Mekelle University, Tigray, Ethiopia, 3 Department of Biostatistics, School of Public health, College of Health Sciences, Mekelle University, Tigray, Ethiopia, 4 Department of Public Health, College of Medicine and Health Sciences, Wachamo University, Sothern Nations Nationalities and Peoples of Ethiopia, Tigray, Ethiopia, 5 Department of Nursing, College of Medicine and Health Sciences, Adigrat University, Tigray, Ethiopia, 6 Department of Public Health, College of Medicine and Health Sciences, Aksum University, Tigray, Ethiopia, 7 Department of Midwifery, College of Medicine and Health Sciences, Adigrat University, Tigray, Ethiopia

* menasebgh@gmail.com

**Data Availability Statement:** All the data related to this research article are available with in the

## Abstract

### Background

Anemia is a global public health problem; but its burden is disproportionately borne among children in the African Regions. The 2016 Ethiopia Demographic and Health Survey report showed that the prevalence of anemia among children 6–59 months of age was 57%; far exceeding the national target of 25% set for 2015. Although studies have been conducted in Ethiopia, multilevel analysis has rarely been used to identify factors associated with anemia among children. Therefore, this study aimed to identify individual and community-level factors associated with anemia among children 6–59 months of age by fitting a multilevel logistic regression model.

### Methods

The data was obtained from the 2016 Ethiopia Demographic and Health Survey, conducted from January to June 2016, and downloaded from the website http://www.DHSprogram.com. The sample was taken using two-stage stratified sampling. In stage one, 645 Enumeration Areas and in stage two 28 households per Enumeration Area were selected. A sample of 7790 children 6–59 months of age was included. Data were analyzed using STATA version 14. A multilevel logistic regression model was fitted and an adjusted odds ratio with a 95% confidence interval was obtained.

manuscript and its supporting information files. However, to replicate our study findings, the EDHS datasets are in the public domain on the DHS measure survey web site which is available at: https://dhsprogram.com/data/available-datasets.cfm. The authors had no special access privileges to the DHS data, and other researchers will be able to access the data in the same manner as the authors using the provided URL link. Data access requests may also be sent to Bridgette Wellington (Data Archivist at The Demographic and Health Surveys (DHS) Program) at E-mail address: archive@dhsprogram.com.

**Funding:** The author(s) received no specific funding for this work.

**Competing interests:** The authors have declared that no competing interests exist.

## Result

From the individual-level factors, anemia was associated most strongly with child age, wealth index, maternal anemia and child stunting followed by child underweight, child fever and birth order whereas from the community-level, the strongest odds of anemia occurred among children from Somali, Harari, Dire Dawa and Afar region followed by Oromia and Addis Ababa. Low community-poverty is a protective factor for anemia. The odds of anemia were 0.81 (95% CI: 0.66, 0.99) times lower for children who were living in communities of lower poverty status than children who were living in communities of higher poverty status. Children from Somali and Dire Dawa had 3.38 (95% CI: 3.25, 5.07) and 2.22 (95% CI: 1.42, 3.48) times higher odds of anemia, respectively than children from the Tigray region.

## Conclusions

This study shows that anemia among children 6–59 months of age is affected both by the individual and community level factors. It is better to strengthen the strategies of early detection and management of stunted and underweight children. At the same time, interventions should be strengthened to address maternal anemia, child fever and poverty, specifically targeting regions identified to have a high risk of anemia.

## Introduction

Globally, anemia affects around 800 million children and women; of which 42.6% are children [1]. African countries have the highest prevalence (62.3%) followed by South-East Asia (53.8%) and Eastern Mediterranean Region (48.6%) [1]. Furthermore, it is a major public health problem in Sub-Saharan African countries with high national prevalence estimated to be above 40% [2].

According to the 2011 Ethiopia Demographic and Health Survey (EDHS), anemia prevalence among children 6–59 months of age was 44% [3]. In addition, the 2016 EDHS report showed that prevalence of anemia among children 6–59 months of age was 57% [4]. This indicates that anemia is a severe public health problem which increased by 13% within five years period, during a period when the government of Ethiopia undertook efforts such as vitamin 'A' supplementation, deworming, and use of fortified foods to reduce anemia through national nutrition programs [5].

Anemia is an important indicator of poor nutrition and health with major consequences of socioeconomic development [6]. Children younger than two years of age with severe anemia are at increased risk of mortality and, even mild forms, which might be corrected cause permanent cognitive damage by decreasing attention span and shortening of memory [7].

Although biochemical and hematological tests exist, hemoglobin concentration in the blood is the most reliable indicator of anemia at the population level [8]. This latter method was used to assess for anemia among children aged 6–59 months in the 2016 Ethiopia Demographic Health Survey (EDHS) [4].

According to the WHO criteria, anemia in children 6–59 months of age is defined as hemoglobin concentration in the blood below 11g/dl [9]. Anemia is said to be a severe public health problem when its prevalence is 40% or more, a moderate public health problem when its prevalence is between 20 and 40% and a mild public health problem when its prevalence is between 5 and 20% in any group [10].

In Ethiopia, previous studies have linked anemia to factors such as child age, child nutritional status, parents' educational level, and wealth index [11–15]. However, almost all studies used single-level analysis techniques with population groups localized in a specific study area [11, 12, 14, 15]. The Single-level analysis assumes that there is no community effect beyond the characteristics of individuals [16]. That is, the impact of community-level factors on anemia among children aged 6–59 months remains under-studied. Moreover, analyzing hierarchical data like the DHS using single-level analysis leads to incorrect estimation of parameters and standard errors [17].

Using multilevel analysis technique, community-level effects can be identified from individual-level effects [18–21]. However, this approach has rarely been used in Ethiopia to identify factors associated with anemia among children. One study that used multilevel-analysis technique failed to examine the effect of some factors such as poor nutritional status of children (stunting, wasting and underweight), child health-related factors (fever, diarrhea and respiratory infection), maternal anemia, and variables aggregated at the community level [18].

Finally, the aims of this study were:

1. To identify individual level factors associated with anemia among children 6–59 months of age in Ethiopia.

2. To identify community level factors associated with anemia among children 6–59 months of age in Ethiopia.

## Methods and materials

### Data source

Data were extracted from the nationally representative 2016 EDHS. The 2016 EDHS is the fourth survey which is implemented by the Central Statistical Agency (CSA) in collaboration with the Ethiopian Ministry of Health under the technical assistance of International Classification of Functioning, Disability, and Health (ICF) through the DHS Program. Ethical approval was obtained from Mekelle University, College of Health Sciences Ethical Review Committee (ERC). Approval to access the 2016 EDHS data set was obtained from DHS Program, after making a request via DHS program website (http://www.DHSprogram.com). The EDHS data has no individual identifiers which could affect the confidentiality of participants and the data was used for analysis purpose only.

**Study population.** Children 6–59 months of age who were living in selected enumeration areas.

### Inclusion and exclusion criteria

The inclusion criteria were children 6–59 months of age who live in the selected enumeration areas (community). And Exclusion criteria were children 6–59 months of age who have no hemoglobin test result.

### Study design and sample size

A population-based cross-sectional survey was used to collect the 2016 EDHS data. The 2016 EDHS had used a stratified two-stage cluster sampling design. Stratification was achieved by separating each region into urban and rural areas, yielding 21 sampling strata. In the first stage, 645 Enumeration Areas (EAs) or clusters were selected. Among the selected 645 EAs, 202 were in urban and 443 in rural areas. In the second stage, households were the sampling

units and a fixed number of 28 households per each EA were selected. From the total of 10,641 under-five years old children, 9504 were children 6–59 months of age. Data on hemoglobin level from the survey were available for 7790 children.

## Definitions of study variables

We assessed the impact of individual and community-level variables on anemia among children 6–59 months of age. We defined anemia in 6–59 month age children as hemoglobin <11 g/dL according to WHO criteria [8]. Individual level variables were: sex, age, birth order, birth weight, religion, number of under-five children, childhood wasting, underweight, stunting, symptoms of acute respiratory infection, child fever and diarrhea, maternal anemia and age, parents' educational and employment status, wealth index, source of drinking water, and type of toilet facility, whereas community-level variables were: region, community-poverty, community-women education and community- women unemployment.

Wealth index is a composite measure of a household's cumulative living standard. It was calculated based on household ownership of selected assets such as televisions and bicycles, cars; materials used for the housing construction; source of drinking water; and type of sanitation facilities. It was then generated using principal components analysis and the individual households were placed on a continuous scale of relative wealth. In the EDHS all mothers and children were assigned a standardized wealth index score. It was measured as a composite variable made up of five quintiles as poorest, poorer, middle, richer and richest [4].

Anthropometrics: stunting was defined as height or length for age (HFA) <-2SD (standard deviation), wasting as weight-for-height (WFH) <-2 SD and underweight as weight-for-age (WFA) <-2 SD.

The community-level variables that directly measure the community characteristics in the 2016 EDHS were the place of residence (rural or urban) and region (either of the nine regions or the two administrative cities). We created also other additional variables by aggregating the individual level's characteristics within their respective clusters. These variables were: community poverty, community women education and community women unemployment.

Community-poverty is the proportion of mothers who reside in poor or poorest households in the community. The aggregate of the poorest or poor individuals can show the overall poverty of the cluster. For this proportion, the median value was calculated as summary statistics and categorized as 'More poverty' or 'less poverty' based on this national median value.

Community-women education (CWE) is proportion of mothers aged 15–49 with secondary or higher education in the community. The median value was calculated as summary statistics that represent the educational status of women in the cluster and was categorized as 'High' or 'Low' based on the national median value.

Community-women unemployment status (CWUe) is proportion of mothers aged 15–49 who were not employed in the community in the past twelve months. It was categorized as high if clusters had more than or equal to the national median proportion of unemployed mothers or low otherwise.

## Methods of data analysis

Before doing any analysis, sampling weight and normalization were done for the sample in order to ensure the representativeness of the sample to different regions and their place of residence. Data were analyzed by Stata version 13 and a multilevel binary logistic regression model was fitted. Frequencies, percentages, graphs and charts were used to describe categorical variables. The effect of each predictor variable on the outcome variable was checked at a significance level of $p \leqq 0.25$ independently [22]. Variables that are statistically significant at the

bivariate multilevel logistic regression analysis were considered as candidates for multivariable analysis. Accordingly, in the multivariable analysis the following variables were adjusted and controlled: number of <5 children in the household, child age, religion of mother (caretaker), birth order, parents employment status and educational level, maternal age, type of toilet facility, source of drinking water, wealth index, child stunting, wasting, underweight, fever, diarrhea, child deworming, symptoms of acute respiratory infection, maternal anemia, community- women education, place of residence, community-poverty, region and community- women unemployment. Adjusted Odds Ratio (AOR) with 95% Confidence Interval (CI) at a significance level of p<0.05 was estimated. The result of multivariable analysis for individual and community-level factors associated with anemia among children aged 6–59 months is shown in Table 3.

Assuming varying intercepts across communities (clusters) but fixed coefficients, four models were developed. The first one was the null model which is fitted without independent variables; this was used to determine the variance in anemia status between the clusters in the sample. Whereas, model one was adjusted for individual-level factors and used to examine their contribution to the variation of anemia status. Model two was adjusted for community level factors and was used for examining whether the community-level variables explain between-cluster variation on childhood anemia. Model three was developed by combining both the individual and community level variables.

**Null model.** For individual $i$ in community j, the model can be represented as [17, 23]:

$$Y_{ij} = \Upsilon_{00} + u_{0j} + \varepsilon_{ij} \ldots \ldots \ldots \ldots \text{null model}$$

Where:

$Y_{ij}$ is anemia status of $i^{\text{th}}$ child in the $j^{\text{th}}$ cluster

$\Upsilon_{00}$ = is the intercept; that is the probability of having anemia in the absence of explanatory variables

$u_{0j}$ = community-level effect; $\varepsilon_{ij}$ error at individual level

**Mixed model.** This model was derived by mixing both individual and community level factors simultaneously [24].

$$Y_{ij} = \Upsilon_{00} + \Upsilon_{k0}X_{kij} + \Upsilon_{0p}z_{pj} + u_{0j} + \varepsilon_{ij} \ldots \ldots$$

Where: The term $\gamma_{k0}$ is the regression coefficient of the individual-level variable $X_k$ and $\gamma_{0p}$ is the regression coefficient of the community-level variable $Zp$. $X_k$ and $Zp$ were individual and community-level explanatory variables respectively. The subscripts $i$ and $j$ represent for the individual level and cluster number respectively.

**Intraclass correlation coefficient (ICC).** A measure of within-cluster homogeneity and the proportion of variance due to between-cluster differences.

$$ICC = \frac{\delta^2_{u_0}}{\delta^2_{u_0} + \pi^2/_3}$$

Where: $\delta^2_{u_0}$ = between cluster (community) variances and $\frac{\pi^2}{3}$ = with in cluster (community) variance. The value of $\frac{\pi^2}{3}$ in case of standard logistic distribution is 3.29 [25].

The null model showed that there was a significant variation in anemia status among clusters ($\delta^2_{u0}$ = 0.76, p-value<0.001). The ICC was 18.77% (95% CI; 0.1598, 0.219), meaning that 18.77% of the total variability in odds of anemia was due to between community differences or attributable to the unobserved factors either at community-level or at individual-level. This indicates that using a multilevel logistic regression model is better for getting valid estimates

than single-level logistic regression [17]. The variance which was due to the clustering effect decreased from 18.77% in the null model to 10.03%, 8.04%, and 7.03% in model one, model two and model three, respectively (S1 Annex).

**Proportional change in variance (PCV).** Calculated with reference to the null model to see relative contribution of factors to explain variation in childhood anemia.

$$PCV = (\frac{\delta^2_{u_0} - \delta^2_{u_1}}{\delta^2_{u_0}}) * 100$$

Where: $\delta^2_{u_0}$ is between community variance in the null model; $\delta^2_{u_1}$ is between community variance in the consecutive models [17].

Model three had the highest PCV which is 67%. This shows that 67% of the variance in the anemia status among children was due to the simultaneous effect of both individual and community-level factors found in the model (S1 Annex).

## Model diagnostics and adequacy checking

Multicollinearity was checked by using a variation inflation factor (VIF) with a cut-off point of 10 [26]. It was checked for the independent variables in the final model and the VIF was found to range from 1.2 to 4.2 with mean VIF of 2.1. This shows that multicollinearity might not be a problem.

Interaction between variables was checked for those variables found significant at the final model. As a result, there were significant interactions between these variable (p<0.05). However, as we examined the interaction effect by fitting regression models that contained interaction terms yields no significant (p>0.05) interaction effect. Model selection was carried out by using Akaike information criteria (AIC). AIC values for each model were compared and the model with the lowest value of AIC was considered as a better explanatory model [25]. Accordingly, model 4 with AIC value of 8281 was selected as the best model for explaining anemia status among children aged 6–59 months in Ethiopia (S1 Annex). Model accuracy was checked by the area under the curve. The Receiver Operating Characteristics curve (ROC-curve) provides a measure of the model's ability to discriminate between those subjects who experience the outcome of interest versus those who do not [27]. The area under the ROC of this model was 0.7376; which means the ROC curve accuracy for outcome variable (anemia) was 74% (Fig 1).

## Results

### Individual-level characteristics of study subjects

From the total of 10,641 under-five years' old children, 9504 were children 6–59 months of age. Data on hemoglobin test result from the survey were available for 7790 children. As a result, 1714 children aged 6–59 month were excluded from the study due to missing data of hemoglobin test result. In addition, the variables of dietary intake and child feeding practices were not included due to missing value. These variables were missing for nearly half of observations.

Seven thousand seven hundred ninety (7790) children 6–59 months of age were included in this study. Above half (52%) of the children were male, and 34.2%, 33.3% and 32.2% where in the age category of 6–23, 24–41 and 42–59 months of age, respectively with mean ± SD (standard deviation) of 32 ±15 months. The prevalence of anemia was 57.6% with a median hemoglobin concentration of 10.7 (IQR: 9.6–11.6).

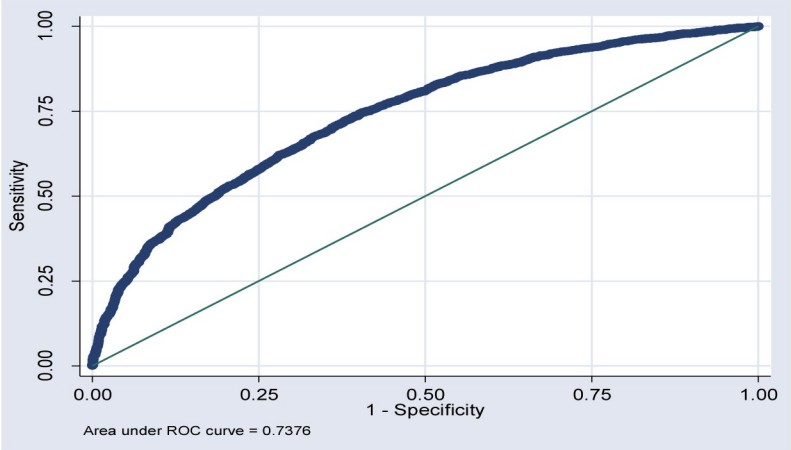

**Fig 1. Receiver operating characteristics curve for anemia of included children age 6–59 months selected from the 2016 EDHS (n = 7790).**

Almost all of the mothers (95%) were living with their respective partners and most of them were Muslims (40%) followed by Orthodox Christians (34%). Nearly half of the mothers (48.7%) were 20–29 years of age. The proportion of no formal education among the children's mothers (67%) was higher than their fathers (48.5%) (Table 1). About one-fifth (23.5%) and 13.4% of respondents fall within the poorest and richest wealth index quintiles, respectively (Fig 2).

Nearly half of the mothers (45.5%) and 92% of husbands were employed in the last twelve months prior to the survey. A quarter (25%) and more than two-fifths of children (41%) were underweight and stunted, respectively. One-third of mothers (30%) were anemic. Fifteen percent of children had fever two weeks prior to the survey. Above half (54.2%) of households had an unimproved type of toilet facility and about 55.5% of households had improved water sources (Table 1).

**Community-level characteristics of study subjects.** About nine in ten (89.9%) of the respondents were rural area residents. Most of the respondents were from Oromia (43.9%) followed by SNNP (21.1%) and Amhara (19.5%) region. Nearly half (47.4%) of the respondents were living in communities with a high proportion of women unemployment. The above half (62.2%) of mothers were from communities with lower poverty status. Above half of mothers (52.5%) were from communities with a low proportion of women education (Table 2).

## Distribution of anemia by individual and community level factors

Under this subtitle, the distribution of anemia by the factors which had significant association with anemia (Table 3) among children aged 6–59 months was elaborated. The highest proportion of anemia (72%) was observed in children 6–23 months old as opposed to the lowest proportion in children 42–59 months old (43%) (Fig 3).

The highest proportion of anemia among children 6–59 months of age was observed in Somali (83%) and Afar (75%) as opposed to the lowest percentage recorded in Amhara (42%) (Fig 4).

Children from anemic mothers (69%) had higher prevalence of anemia as compared to children whose mothers (caretakers) were not anemic (Fig 5).

**Table 1. Individual-level characteristics of included children age 6–59 months selected from the 2016 EDHS.**

| Variable | Frequency (weighted) | Percentage (%) |
|---|---|---|
| **Child Sex** | | |
| Male | 4039 | 51.8 |
| Female | 3751 | 48.2 |
| **Child age (months)** | | |
| 6–23 | 2667 | 34.2 |
| 24–41 | 2597 | 33.3 |
| 42–59 | 2526 | 32.4 |
| **Birth weight** | | |
| Average | 3276 | 42.1 |
| Smaller than average | 2001 | 25.7 |
| Larger than average | 2513 | 32.3 |
| **Birth order** | | |
| 1 | 1362 | 17.5 |
| 2–3 | 2417 | 31 |
| 4–5 | 1870 | 24 |
| > = 6 | 2141 | 27.5 |
| **Religion** | | |
| Orthodox | 2677 | 34.4 |
| Protestant | 1722 | 22.1 |
| Muslim | 3153 | 40.5 |
| Other | 238 | 3.1 |
| **Maternal age** | | |
| 15–19 | 208 | 2.7 |
| 20–29 | 3796 | 48.7 |
| 30–39 | 3065 | 39.3 |
| 40–49 | 721 | 9.3 |
| **Maternal educational level** | | |
| No education | 5224 | 67.1 |
| Primary | 2091 | 26.8 |
| Secondary | 318 | 4.1 |
| Higher | 157 | 2 |
| **Husband educational level** | | |
| No education | 3582 | 48.5 |
| Primary | 3008 | 40.7 |
| Secondary | 526 | 7.1 |
| Higher | 275 | 3.7 |
| **Variable** | **Frequency (weighted)[1]** | **percentage** |
| **Maternal employment status** | | |
| No | 4248 | 54.5 |
| Yes | 3542 | 45.5 |
| **Husband employment status** | | |
| No | 599 | 8.1 |
| Yes | 6792 | 91.9 |
| **Source of drinking water** | | |
| Improved | 4320 | 55.5 |
| Unimproved | 3470 | 44.5 |
| **Types of toilet facility** | | |

(*Continued*)

**Table 1.** (Continued)

| | | |
|---|---|---|
| Improved | 719 | 9.2 |
| Unimproved | 4221 | 54.2 |
| No facility | 2850 | 36.6 |
| **Child wasting** | | |
| No | 7049 | 90.6 |
| Yes | 731 | 9.4 |
| **Child underweight** | | |
| No | 5789 | 74.7 |
| Yes | 1964 | 25.3 |
| **Child stunting** | | |
| No | 4490 | 59 |
| Yes | 3175 | 41 |
| **Maternal anemia** | | |
| No | 5381 | 70 |
| Yes | 2320 | 30 |
| **Child deworming** | | |
| No | 6787 | 87 |
| Yes | 1003 | 13 |
| **Child diarrhea** | | |
| No | 6788 | 87.14 |
| Yes | 1002 | 12.86 |
| **Child fever** | | |
| No | 6605 | 84.79 |
| Yes | 1185 | 15.21 |
| **Symptoms of acute respiratory infection** | | |
| No | 6157 | 79 |
| Yes | 1633 | 21 |

[1] Weighting variable(wgt) = women individual sample weight (v005)/$10^6$; normalization variable(w) = un-weighted sample/weighted sample $^*$wgt.

## Individual and community-level factors associated with anemia

From the individual-level factors, anemia was most strongly associated with child age, wealth index, maternal anemia and child stunting followed by child underweight, child fever and birth order whereas from the community-level, the strongest odds of anemia occurred among children from Somali, Harari, Dire Dawa and Afar region followed by Oromia and Addis Ababa regions (Table 3).

The odds of anemia were 4.45 (95%CI; 3.62, 5.36) and 1.83 (95% CI: 1.59, 2.09) times higher for children 6–23 and 24–41 months of age than children at age 42–59 months, respectively. Children from the poorest families had 1.51 (95% CI: 1.11, 2.04) times higher odds of anemia than children from the richest families. The odds of anemia were 1.26 (95% CI: 1.00, 1.61) times higher for children with birth order six and above than first-order children (Table 3).

Underweight children had 1.34 (95% CI: 1.14, 1.57) times higher odds of anemia than children who were not underweight. The odds of anemia were 1.40 (95% CI: 1.24, 1.59) times higher among children who were stunted than children who are not stunted. Maternal anemia was positively associated with childhood anemia. Children whose mothers were anemic had 1.42 (95% CI: 1.21, 1.55) times higher odds of anemia than children from non-anemic

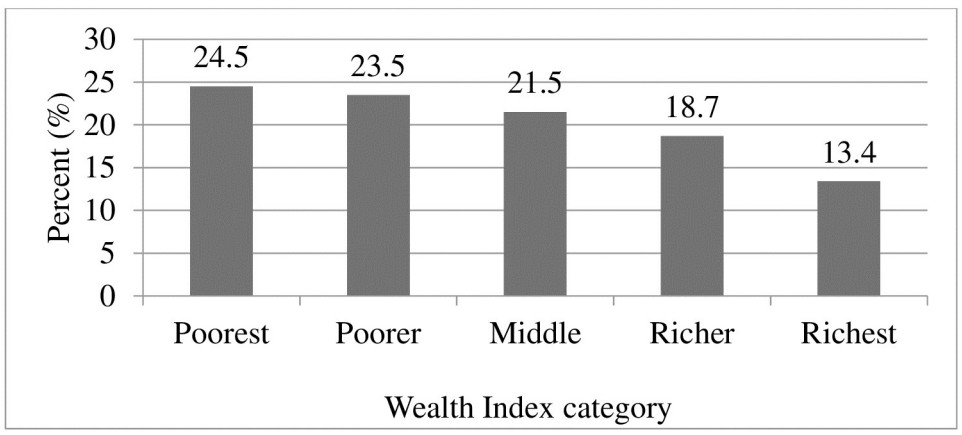

**Fig 2. Wealth index characteristics of children age 6–59 months selected from the 2016 EDHS (n = 7790).**

**Table 2. Community-level characteristics of included children age 6–59 months selected from the 2016 EDHS.**

| Variables | Frequency (weighted) | Percentage (%) |
|---|---|---|
| **Region** | | |
| Tigray | 526 | 6.8 |
| Afar | 76 | 0.98 |
| Amhara | 1520 | 19.5 |
| Oromia | 3418 | 43.9 |
| SNNP[1] | 1637 | 21.1 |
| Benishangul Gumuz | 83 | 1.1 |
| Gambela | 18 | 0.23 |
| Somali | 321 | 4.11 |
| Harari | 15 | 0.2 |
| Addis Abeba | 148 | 1.9 |
| Dire Dawa | 29 | 0.37 |
| **Place of residence** | | |
| Rural | 7002 | 89.9 |
| Urban | 788 | 10.1 |
| **Community-women education[2]** | | |
| Low | 4092 | 52.5 |
| High | 3698 | 47.5 |
| **Community- women unemployment[3]** | | |
| Low | 4098 | 52.6 |
| High | 3692 | 47.4 |
| **Community -poverty[4]** | | |
| Low | 4098 | 62.2 |
| High | 2945 | 37.8 |

[1] Sothern Nations and Peoples of Ethiopia.

[2] Proportion of mothers aged 15–49 with secondary or higher education in the community.

[3] Proportion of mothers aged 15–49 who were not employed in the community in the past twelve months.

[4] Proportion of mothers who reside in poor or poorest households in the community.

**Table 3. Individual and community-level factors associated with anemia among children age 6–59 months selected from the 2016 EDHS.**

| Variables | Anemia Status | | AOR [95%CI] | P-value |
|---|---|---|---|---|
| | Frequency (%) | | | |
| | No | Yes | | |
| **Individual level factors** | | | | |
| **Child age (months)** | | | | |
| 6–23 | 746.5 (28) | 1921 (72) | 4.45 [3.62, 5.36] | 0.000 |
| 24–41 | 1116 (43) | 1481 (57) | 1.83 [1.59, 2.09] | 0.000 |
| 42–59 | 1444 (57) | 1082 (43) | 1 | |
| **Wealth index** | | | | |
| Poorest | 583 (32) | 1245 (68) | 1.51[1.11, 2.04] | 0.035 |
| Poorer | 770 (42) | 1058 (58) | 1.29 [0.98, 1.72] | 0.052 |
| Middle | 772 (46) | 901 (54) | 1.11 [.84, 1.46] | 0.296 |
| Richer | 639 (45) | 776 (55) | 1.19 [.90, 1.55] | 0.106 |
| Richest | 542 (52) | 504 (48) | 1 | |
| **Birth order** | | | | |
| 1 | 602 (44) | 760 (55.8) | 1 | |
| 2–3 | 1053 (43) | 1365 (56.5) | 1.08[.91, 1.28 | 0.112 |
| 4–5 | 779 (42) | 1100 (58.4) | 1.16 [.94, 1.43] | 0.062 |
| > = 6 | 873 (40.8) | 1268 (59.2) | 1.26 [1.00, 1.61] | 0.044 |
| **Child underweight** | | | | |
| No | 2591 (45) | 3198. (55) | 1 | |
| Yes | 701 (36) | 1262 (64) | 1.34 [1.14, 1.57] | 0.000 |
| **Child stunting** | | | | |
| No | 2052 (46) | 2438 (54) | 1 | |
| Yes | 1223(39) | 1953 (61) | 1.4 [1.24, 1.59] | 0.000 |
| **Maternal anemia** | | | | |
| No | 2563 (48) | 2818 (52) | 1 | |
| Yes | 723 (31 | 1597 (69) | 1.42 [1.21, 1.55] | 0.000 |
| **Child fever** | | | | |
| No | 2886 (44) | 3719 (56) | 1 | |
| Yes | 420 (35) | 764 (65) | 1.32[1.09, 1.60] | 0.004 |
| **Community level factors** | | | | |
| **Community- poverty** | | | | |
| More Poverty | 1020 (35) | 1925 (65) | 1 | |
| Less poverty | 2287 (47) | 2558 (53) | 0.81[.66, .99] | 0.032 |
| **Region** | | | | |
| Tigray | 242 (46) | 284 (54) | 1 | |
| Afar | 19 (25) | 57 (75) | 1.66 [1.11, 2.48] | 0.001 |
| Amhara | 873 (57) | 647 (42) | 0.71 [0.52, 0.96] | 0.089 |
| Oromia | 1170 (34) | 2,248 (66) | 1.62 [1.16, 2.26] | 0.000 |
| Somali | 54 (17) | 267 (83) | 3.38 [3.25, 5.07] | 0.000 |
| Benishangul | 47 (57) | 36 (43) | .58 [.40, .82] | 0.000 |
| SNNP | 804 (49) | 833 (51) | 1.00 [0.71, 1.42] | 0.113 |
| Gambela | 8 (43) | 10 (57) | 1.20 [0.81, 2.78] | 0.052 |
| Harari | 5 (33) | 10 (67) | 1.88 [1.23, 2.88] | 0.000 |
| Addis Ababa | 76 (51) | 72 (49) | 1.54[1.01, 2.35] | 0.001 |

(*Continued*)

**Table 3.** (Continued)

| Variables | Anemia Status | | AOR [95%CI] | P-value |
|---|---|---|---|---|
| | Frequency (%) | | | |
| | No | Yes | | |
| Dire Dawa | 8 (28) | 21 (72) | 2.22 [1.42, 3.48] | 0.000 |

Key: 1- reference category, AOR- Adjusted Odds Ratio, CI- Confidence Interval, SNNPR- Southern Nations, Nationalities and People's Region.

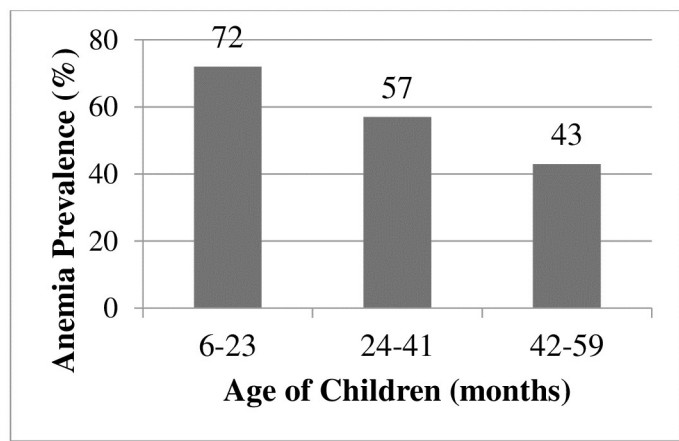

**Fig 3. Anemia distribution by age of children in Ethiopia: A multilevel analysis using EDHS 2016 (n = 7790).**

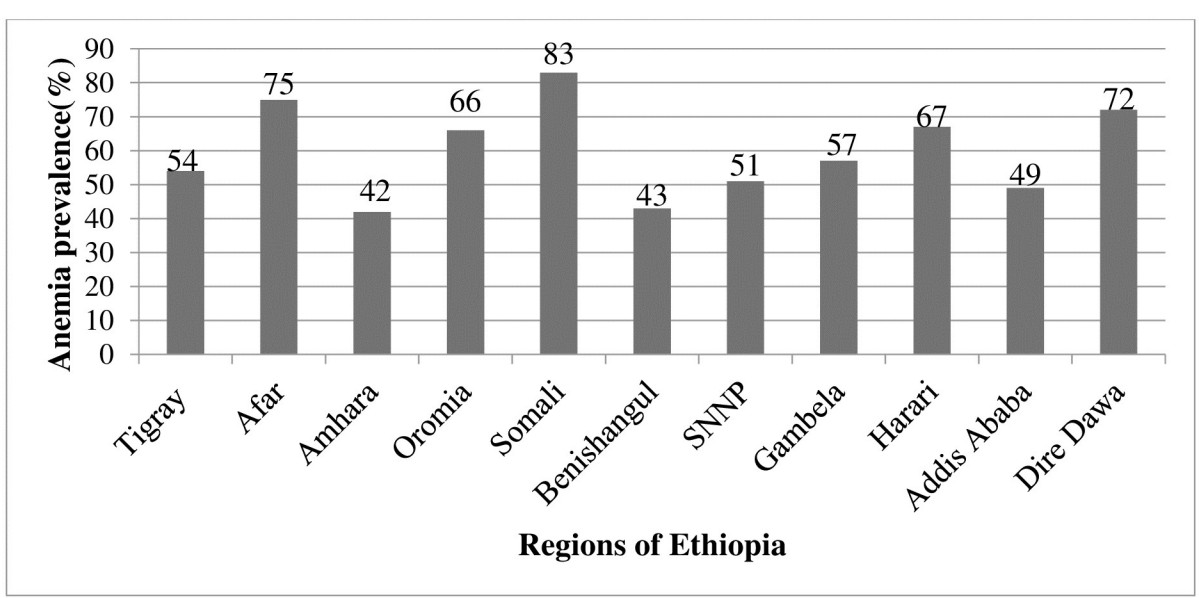

**Fig 4. Percentage distribution of anemia among children 6–59 months by Region of respondents in Ethiopia: A multilevel analysis using EDHS 2016 (n = 7790).**

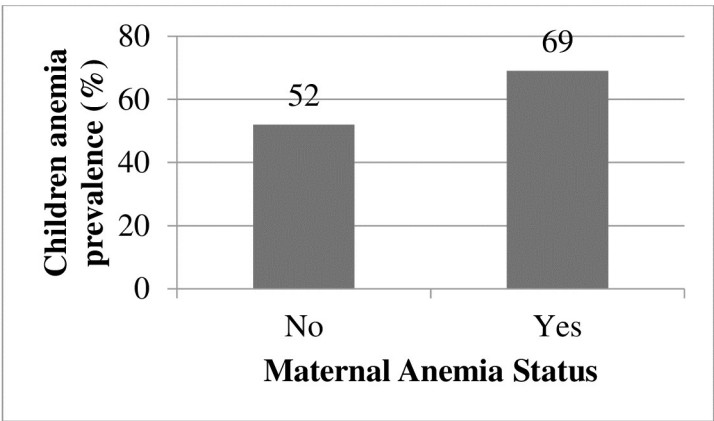

**Fig 5. Distribution of anemia prevalence among children 6–59 months in Ethiopia: A multilevel analysis using EDHS 2016 (n = 7790).**

mothers. The odds of anemia were 1.32 (95% CI: 1.09, 1.60) times higher for children who suffered a fever two weeks prior to the survey than children with no fever (Table 3).

The region was a significant predictor of anemia among children. Children from Somali, Dire Dawa and Harari had 3.38 (95% CI: 3.25, 5.07), 2.22 (95% CI: 1.42, 3.48) and 1.88 (95% CI: 1.23, 2.88) times higher odds of anemia than children, respectively than children from Tigray region. The odds of anemia were 0.71 (95% CI: 0.52, 0.96) and 0.58 (95% CI: 0.40, 0.82) times lower among children who were living in Amhara and Benishangul than children, respectively than children from Tigray region. Children who were living in communities of less poverty status had 0.81 (95% CI: 0.66, 0.99) times lower odds of anemia than others (Table 3).

## Discussion

This study aimed to identify individual and community-level factors associated with anemia among children aged 6–59 months. We found that anemia among children aged 6–59 was most strongly associated with individual-level factors such as child age, wealth index, maternal anemia and child stunting followed by child underweight, child fever and birth order, whereas from the community-level the strongest odds of anemia occurred among children from Somali, Harari, Dire Dawa and Afar region followed by Oromia and Addis Ababa. This was also supported by the observed heterogeneity in odds of anemia between communities.

Child age was negatively associated with anemia in which the odds of anemia decreased as the age of child increased. Children 6–23 months old had higher odds of anemia as compared to children 42–59 months old. This result is in line with some previous studies done in Bangladesh and Ethiopia [13, 18, 28]. This might be due to children experiencing intense growth and development in the first 2 years of life, resulting in a high demand for iron [29]. Additionally, complementary foods are often initially rejected by the infant, thereby exacerbating the risk of anemia.

Children with six and above birth order had higher odds of anemia than first-order children. This result is similar to findings from a prior study done in New Delhi, India [30]. This could be due to increasing birth order might relate to maternal depletion of iron; as the finding of this study showed that maternal anemia leads to child anemia.

Children from the poorest households had higher odds of anemia than children from richest households. This finding is in agreement with what was reported in Bangladesh, Malawi and Ethiopia [11, 21, 28]. This could be explained as poor families are less likely to afford adequate and diversified foods and access to health care which leads to poor child health outcomes.

The odds of anemia were higher for children from anemic mothers than non-anemic mothers. This is supported by studies conducted in India and Ethiopia [13, 31]. The reasons may be mothers and children share common home environments, socioeconomic, and dietary conditions. Moreover, maternal iron deficiency is associated with low birth weight; even children born with adequate weight have reduced iron reserves when their mothers are anemic [32].

This study also revealed that, being underweight or stunted was positively associated with anemia. The association between these anthropometric indices and anemia has been observed in other studies [13, 15, 21, 28]. The possible explanation could be stunting, underweight and anemia are all caused by malnutrition, and thus follow a similar causal pathway that is; feeding children less than four times a day and low dietary diversity. Another explanation could be nutritional inadequacies may impair immunity with a repeated infection which, in turn, depletes iron stores.

The presence of fever in the last two weeks prior to the survey was found to be a significant determinant of anemia. This finding confirms the findings from Indonesia, Burma, Nigeria and Malawi [21, 33–35]. This could explain as fever is a symptom of acute febrile illness such as malaria; which might cause red blood cell destruction. Inflammation also decreases red blood cell production [36]. Another explanation could be that sick children are known to have poorer appetites; hence a lower dietary intake.

This study found that parents' educational level had no significant association with anemia among children aged 6–59 months. This finding is in contrast with several other studies where an association was observed, especially with maternal educational level [18, 28, 35]. The possible explanation could be controlling socio-economic factors such as parents' employment, wealth-index, and the composite factors like community poverty in the model may have a more pronounced effect in determining childhood anemia than educational level.

Region was found to be significantly associated with anemia. This finding is supported by other studies from Ghana and Ethiopia [18, 37]. Children from Somali, Dire Dawa, Harari and Afar regions had higher odds of anemia than children from Tigray. This could be because of differences in living standards, socioeconomic status and cultural norms regarding feeding habits among regions. People who are living in Afar and Somalia make their living from livestock production and the main daily meal is milk. A previous study documented that milk reduces the bioavailability of iron which leads to anemia [38]. Another finding of this study was the significant association of community poverty status and anemia among children. Children who live in communities with high poverty status had higher odds of anemia than others. This significant association might be due to less access to health care, fewer job opportunities and lack of other social services within the community which resulted in lower-income.

## Limitations

The limitations of this study include: not controlling variables of dietary intake and child feeding practices due to missing values. Other possible child-related explanatory variables such as parasitic infection and chronic illness were not included in the analysis because these variables are not in the EDHS data. The data based on self-reporting are limited by recall and misclassification biases, and only children living at the time of the survey were included. In addition, the community variability in the combined model was 7.03%. This indicates that there are still

other variables that are not controlled. These variables could be parasitic infection, immunization history, chronic infections, dietary intake and child feeding practices.

## Conclusion

We found that anemia among children aged 6–59 was most strongly associated with individual-level factors such as child age, wealth index, maternal anemia and child stunting followed by child underweight, child fever and birth order whereas from the community-level the strongest odds of anemia occurred among children from Somali, Harari, Dire Dawa and Afar region followed by Oromia and Addis Ababa region. Interventions like community-based screening for early detection and management of stunted and underweight children should strengthen to reduce anemia. As our model predicts, wealth index has additional contribution to anemia in Ethiopian children 6–59 months. This suggests that interventions targeted to the improvement of Economic subsidy may contribute to a reduction in childhood anemia and its devastating complications. In addition, special attention should be given to children less than two years of age. Similarly, priority should be given to regions such as Somali, Harari and Afar during the implementation of interventions to reduce anemia.

## Supporting information

**S1 Annex. Random effects estimates of anemia for included children age 6–59 months selected from the 2016 (n = 7790).**
(DOCX)

**S1 Dataset. Child dataset final.**
(DTA)

**S1 File.**
(PDF)

## Acknowledgments

The authors would like to thank the Department of Public Health, Adigrat University, and Mekelle University, College of Health Sciences, Tigray, Ethiopia. The authors also acknowledge EDHS, DHS program and ICF International for providing us permissions to access the data set.

## Author Contributions

**Conceptualization:** Menaseb Gebrehaweria Gebremeskel.

**Data curation:** Menaseb Gebrehaweria Gebremeskel.

**Formal analysis:** Menaseb Gebrehaweria Gebremeskel, Haftay Gebremedhin.

**Funding acquisition:** Menaseb Gebrehaweria Gebremeskel.

**Methodology:** Menaseb Gebrehaweria Gebremeskel, Afework Mulugeta, Abate Bekele, Lire Lemma, Muzey Gebremichael, Haftay Gebremedhin, Berhe Etsay, Fre Gebremeskel, Selam Shushay.

**Validation:** Muzey Gebremichael.

**Writing – original draft:** Menaseb Gebrehaweria Gebremeskel.

**Writing – review & editing:** Menaseb Gebrehaweria Gebremeskel, Afework Mulugeta, Abate Bekele, Lire Lemma, Muzey Gebremichael, Haftay Gebremedhin, Berhe Etsay, Tesfay Tsegay, Yared Haileslasie, Yohannes Kinfe, Fre Gebremeskel, Letemichael Mezgebo, Selam Shushay.

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
