## [Decision Letter · Decision Letter 0]

14 Jul 2020

PONE-D-20-09803

INDIVIDUAL AND COMMUNITY LEVEL FACTORS ASSOCIATED WITH ANEMIA AMONG CHILDREN 6 - 59 MONTHS OF AGE IN ETHIOPIA: A FURTHER ANALYSIS OF 2016 ETHIOPIA DEMOGRAPHIC AND HEALTH SURVEY

PLOS ONE

Dear Dr. Gebremeskel,

Thank you for submitting your manuscript to PLOS ONE. After careful consideration, we feel that it has merit but does not fully meet PLOS ONE’s publication criteria as it currently stands. Therefore, we invite you to submit a revised version of the manuscript that addresses the points raised during the review process.

We look forward to receiving your revised manuscript.

Kind regards,

Aamer Imdad

Academic Editor

PLOS ONE

Journal Requirements:

2. Please include a copy of Table 3 which you refer to in your text

Additional Editor Comments (if provided):

Please review the comments from reviewer 1. We dont need to change the language from past to present tense.

Reviewers' comments:

Reviewer's Responses to Questions

**Comments to the Author**

1. Is the manuscript technically sound, and do the data support the conclusions?

Reviewer #1: Yes

Reviewer #2: Yes

2. Has the statistical analysis been performed appropriately and rigorously? 

Reviewer #1: Yes

Reviewer #2: Yes

3. Have the authors made all data underlying the findings in their manuscript fully available?

Reviewer #1: Yes

Reviewer #2: Yes

4. Is the manuscript presented in an intelligible fashion and written in standard English?

Reviewer #1: Yes

Reviewer #2: Yes

5. Review Comments to the Author

Reviewer #1: INDIVIDUAL AND COMMUNITY LEVEL FACTORS ASSOCIATED WITH ANEMIA AMONG CHILDREN 6 - 59 MONTHS OF AGE IN ETHIOPIA: A FURTHER ANALYSIS OF 2016 ETHIOPIA DEMOGRAPHIC AND HEALTH SURVEY

General comments:

The information you have analyzed and interpreted is valuable. I have made suggestions to improve the organization and presentation of your data.

Cut down on stating data and use the space to provide more interpretation. For example, use more graphics (ex. bar graph) showing the distribution of anemia in your strongest predictors.

Review

Abstract

Background:

Result

- Consider wording the results as Anemia was associated most strongly with ….

Conclusions

- Consider using present tense throughout the paper (e.g. “shows” not showed)

- Malnutrition is not specifically discussed in your results, why is it one of the main conclusion?

Introduction

- “below” not “bellow”

- Childhood anemia is defined by age group and gender. I disagree with your definition. Anemia in children 6-59 months specifically is defined by WHO as <11g/dL hgb. Maybe that is what you meant? I would suggest stating in your methods that you defined anemia as hemoglobin <11g/dL based on xxxx definition from the WHO. For

- I would suggest starting with sentence on prevalence of anemia in children worldwide versus in Ethiopia, why it is important/what the complications are, then why hemoglobin is used as a measure of anemia, then the WHO definition of anemia in children 6-59 months of age, and then discuss more details on Ethiopian anemia classification and problems specific to anemia.

- Specifically state the aims. The aims of this study are to 1), 2), 3)….

Methods

- Where are your inclusion and exclusion criteria? What is your study population?

- I would remove all formulas from your methods unless they are novel or controversial methods of analysis.

-

Data Source

- I would again spell out EDHS the first time in this section, but that is up to you.

- Which MOH? Specify Ethiopian Ministry of Health.

- Yes, we need to know that you have permission to access the EDHS database, but I don’t think we need to know how you obtained permission.

- “through” not “thorough”

Study Design and Sample Size

- The cross sectional survey you describe is the original method of the EDHS data collection? Or are we talking about your study?? If this is your study, then very briefly describe the methods of the EDHS in the first paragraph.

- We need your inclusion and exclusion criteria in order to understand why it is important that 9504 children were 6-59 months of age.

- Revise grammar for the sentence “the 2016 EDHS was used…”

- Please explain the rationale for the use of clusters if this is your selection methodology

-

Study Variables (Maybe a better title is Definitions?)

- What do you mean the variables were selected from the literature?

- I would not go to the extent of telling us if your variables are dichotomous.

- Perhaps simplify this paragraph by first stating that “We assessed the impact of individual and community-level variables on our primary outcome of anemia. We defined anemia as … (see below). Individual level variables were …. Community-level variables were …..”

- “We defined anemia in 6-59 month children as <11 g/dL according to WHO criteria (reference).” Please see earlier comments. Remember that unless you specify, I don’t even know that you are discussing children 6-59 months only.

- Definitions are important. Consider using full sentences such as, “Wealth Index is a composite measure…” instead of : .

- Consider combining anthropometrics into a paragraph “Anthropometrics: Stunting was defined as height or length for age, HFA) <-2, wasting as ….

- What about children without a mother? Are they included? I wonder if some of the “mothers” are other relatives who are the child’s guardian.

- What do you mean by “poor” or “poorest” etc.? Are these economic quintiles? Please define objectively this subjective term. This makes it difficult for me to interpret Figure 2.

Methods of data analysis

- I would cut this section down to two paragraphs. The explanation of your model also needs to be condensed to one or two paragraphs. You may consider sharing your variable selection as an Appendix or supplement.

Results

- How many children were included?

- Please first state how many children you included and basic demographics (age, gender). Then state prevalence of anemia. Then go on to other variables.

- The sentence tells me that children were living with their partners. Surely this is not what is meant. Consider using “caretakers” to clarify when you discuss characteristics of the caretaker or mother.

- I would be more interested in seeing anemia sorted by economic status than a pie chart only of economic status. I think seeing child anemia sorted by maternal anemia (what is the definition of this, by the way?) would also provide useful information, as maternal anemia is a known risk factor for childhood anemia especially for the youngest.

- Table 2. I think this should be Table 1. I would not present the current Table 1. Good to have this table (2), but some of this information can be presented as supplemental data. Please indicate the units, for example child age (months). Please consider labeling this as something like “Table x. Demographic characteristics of included children age 6-59 months selected from the 2016 EDHS.”

- Why does birth order 4-5 matter more than >=6?

- Community Level characteristics table: Please note this is mislabeled as Table 1. Please also see above suggestion for the title.

Individual and community-level factors associated with anemia

- Consider presenting the odds ratios in Table 1 next to the demographics. This would condense a lot of information into less space. Alternatively, cut down your demographics to the essential for Table 1 (currently labelled 2) and then keep Table 4 as is. Then present some of your data in graphs as well.

- Consider instead of simply stating all the OR, giving some context of importance. For example, “the strongest odds of anemia occurred with x and y (data).” Or “anemia was most strongly associated with ….”

- Table 4: I would remove crude OR and present the adjusted OR with p value in the next column. Also, see above to cut down on demographics table as much of the same information is repeated here. Please explain what you adjusted for. Tell me more about the results of your multivariate model? Which variables made it into your model? This is not clear.

Discussion

- “This study aimed to ….” (not was aimed to)

- “This finding indicated that” is too specific to one finding when you are discussing an entire study. Consider wording like: “We found that anemia was most strongly associated with … “ or “We found that … were the strongest predictors of anemia followed by…”

- Consider making a figure using a flowchart of the strongest predictors of anemia (from your model) with the adjusted OR or other information along the arrows that lead to anemia. This would be an excellent visual representation of your most important conclusion.

- What about confounders or collinearity? How confident are you that those factors did not influence your model?

Conclusion

- I would emphasize here the strongest of all predictors first.

- I think going from economic quartile as a risk factor to suggesting food subsidies is a big stretch. Perhaps say something like “our model predicts that … are most strongly associated with anemia in Ethiopian children 6-59 months with additional contribution from …. This suggests that interventions targeted to the improvement of …. may contribute to a reduction in childhood anemia and its devastating complications.”

Reviewer #2: This study addressed an important question. The study question is clearly defined and methods are described in detail and the interpetation of results is appropriate. This study will be an important contribution to the existing literature about risk of anemia in young children.

6. PLOS authors have the option to publish the peer review history of their article (what does this mean?). If published, this will include your full peer review and any attached files.

Reviewer #1: No

Reviewer #2: No

---

## [Author Response · Author response to Decision Letter 0]

12 Aug 2020

Responses to reviewers 

Reviewer #1

Dear reviewer, thank you for your thorough review of our manuscript. Your comments are constructive and very good lessons for us. We have tried to put our responses to your constructive comments and questions. The responses are put immediately after the questions, suggestions or comments. 

General comments:

The information you have analyzed and interpreted is valuable. I have made suggestions to improve the organization and presentation of your data.

1. Cut down on stating data and use the space to provide more interpretation. For example, use more graphics (ex. bar graph) showing the distribution of anemia in your strongest predictors

As per the comment, we presented the distribution of anemia by the factors found to have significant association using pie chart and bar graphs.

Review

2.Abstract

Background:

Result

2.1. Consider wording the results as Anemia was associated most strongly with ….

The result part of the abstract is reviewed and as per the comments given, we rewrite it as: Anemia was associated most strongly with child age, stunting, underweight, fever, maternal anemia, low poverty-community and Region. The odds of anemia were 0.81 AOR=0.81; 95% CI: 0.66, 0.99) times lower for children who were living in communities of lower poverty status than children who were living in communities of higher poverty status. Children from Somali and Dire Dawa had 3.38 (AOR=3.38; 95% CI: 3.25, 5.07) and 2.22 (AOR=2.22; 95% CI: 1.42, 3.48) times higher odds of anemia, respectively than children from the Tigray Region. 

Conclusions

2.2. Consider using present tense throughout the paper (e.g. “shows” not showed)

The conclusion is reviewed not only in the abstract but also in the main body, and based on your comment we rewrite it as: This study shows that childhood anemia is affected both by the individual and community level factors. It is better to strengthen the strategies of early detection and management of stunted and underweight children. At the same time, interventions should be strengthened to address maternal anemia, child fever and poverty, specifically targeting regions identified to have a high risk of anemia. 

2.3. Malnutrition is not specifically discussed in your results, why is it one of the main conclusion? is reviewed not only in the abstract but also in the main body

Yes. Malnutrition itself is not discussed as our result. However, we found that child stunting and underweight had strong association with anemia, and these variables are some forms of Malnutrition in children. But now your comment reminds us it is better to conclude based on our result i.e stunting and underweight. So, we rewrite the conclusion as: This study shows that childhood anemia is affected both by the individual and community level factors. It is better to strengthen the strategies of early detection and management of stunted and underweight children. At the same time, interventions should be strengthened to address maternal anemia, child fever, food insecurity and poverty, specifically targeting regions identified to have a high risk of anemia. 

3. Introduction

3.1. “below” not “bellow”

The word bellow is deleted. And the sentence now becomes: Childhood anemia is defined as hemoglobin concentration in the blood below 11g/dl.

3.2. Childhood anemia is defined by age group and gender. I disagree with your definition. Anemia in children 6-59 months specifically is defined by WHO as <11g/dL hgb. Maybe that is what you meant? I would suggest stating in your methods that you defined anemia as hemoglobin <11g/dL based on xxxx definition from the WHO. For

We defined anemia in 6-59 month age children as hemoglobin <11 g/dL according to WHO criteria.

3.3. I would suggest starting with sentence on prevalence of anemia in children worldwide versus in Ethiopia, why it is important/what the complications are, then why hemoglobin is used as a measure of anemia, then the WHO definition of anemia in children 6-59 months of age, and then discuss more details on Ethiopian anemia classification and problems specific to anemia.

We carefully reviewed the introduction, and we rewrite it based on your suggestion in the revised document.

3.4. Specifically state the aims. The aims of this study are to 1), 2), 3)….

As per the comment, the aims of this study are :

1. To identify individual level factors associated with anemia among children 6-59 months of age in Ethiopia.

2. To identify community level factors associated with anemia among children 6-59 months of age in Ethiopia

4. Methods

4.1. - Where are your inclusion and exclusion criteria? What is your study population?

Based on your comment, we included the inclusion and exclusion criteria as follows. for general information, the EDHS kid record file is source of our sample, and this data contains under-five children (children 0-59 months of age). 

Inclusion criteria: children 6-59 months of age who live in the selected enumeration areas (community).

Exclusion criteria: children 6-59 months of age who have no hemoglobin test result. So, according to this inclusion and exclusion criteria, we have had a total of 10,641 under-five children regardless of their hemoglobin test result. Then from these children, we exclude 1137 children less than 6 months of age (10641-1137=9504 children 6-59 months age regardless of their hemoglobin test result). Then from these children we exclude again 1714 children 6-59 months age who have no hemoglobin test result. Finally, we consider 7790 children 6-59 months age as eligible for our study. 

The study population is children 6-59 months of age who were living in selected enumeration areas.

4.2. I would remove all formulas from your methods unless they are novel or controversial methods of analysis.

As mentioned in the method section of the document, the analysis method we applied is multilevel binary logistic regression analysis. These multilevel models are different from the ordinary single level logistic regression in such a way that they can handle hierarchical nature of data like that of the Demographic and Health Survey (DHS) data. Recently, these models are becoming popular models. However, in our country Ethiopia, these models are still rarely used. That is why we preferred to include their formula in our document. 

Data Source

4.3. I would again spell out EDHS the first time in this section, but that is up to you.

EDHS is abbreviation for the Ethiopia Demographic and Health Survey and mostly it is the name given for the respective Ethiopia Demographic Health Surveys conducted in Ethiopia. 

4.4. Which MOH? Specify Ethiopian Ministry of Health.

We correct it and the sentence becomes “The 2016 EDHS is the fourth survey which is implemented by the Central Statistical Agency (CSA) in collaboration with the Ethiopian Ministry of Health under the technical assistance of International Classification of Functioning, Disability, and Health (ICF) thorough the DHS Program”.

4.5. Yes, we need to know that you have permission to access the EDHS database, but I don’t think we need to know how you obtained permission.

Our intention to write how we obtained permission to access the EDHS data was just to convince readers of this paper that we are legal. But based on your comment we removed the sentences how we obtained permission.

4.6. “through” not “thorough”

We correct it.

4.7. Study Design and Sample Size. The cross sectional survey you describe is the original method of the EDHS data collection? Or are we talking about your study?? If this is your study, then very briefly describe the methods of the EDHS in the first paragraph.

The cross sectional survey we described in our document is the original method of the EDHS data collection. We did nothing on the study design and sampling design just we used the original method of the EDHS data collection methods.

4.8. We need your inclusion and exclusion criteria in order to understand why it is important that 9504 children were 6-59 months of age. 

Inclusion criteria: children 6-59 months of age who live in the selected enumeration areas (community).

Exclusion criteria: children 6-59 months of age who have no hemoglobin test result. So, according to this inclusion and exclusion criteria, we have had a total of 10,641 under-five children regardless of their hemoglobin test result. Then from these children, we exclude 1137 children less than 6 months of age (10641-1137=9504 children 6-59 months age regardless of their hemoglobin test result). Then from these children we exclude again 1714 children 6-59 months age who have no hemoglobin test result. Finally, we consider 7790 children 6-59 months age as eligible for our study”.

4.9. Revise grammar for the sentence “the 2016 EDHS was used…”

We corrected it as: The 2016 EDHS had used a stratified two-stage cluster sampling design.

4.10. Study Variables (Maybe a better title is Definitions?)

We amend it as: Definitions of study variables

4.11. What do you mean the variables were selected from the literature?

During the conception of our study, we reviewed literatures just to know what was done before regarding anemia among children 6-59 months age and what is left to solve the problem. So, during this we observed and identified variables included in the literatures done before. Bay the way, we include also new variables for the first time in our study. 

4.12. I would not go to the extent of telling us if your variables are dichotomous.

Yes you are right. But we don’t think that all readers have the same knowledge in understanding the outcome variable (anemia) whether it is binary variable. That is why we indicated as it is dichotomous. 

4.13. Perhaps simplify this paragraph by first stating that “We assessed the impact of individual and community-level variables on our primary outcome of anemia. We defined anemia as … (see below). Individual level variables were …. Community-level variables were …..”

We reviewed this paragraph and based on your comment, we rewrite it as: 

We assessed the impact of individual and community-level variables on anemia among children 6-59 months of age. We defined anemia in 6-59 month children as hemoglobin (hg) <11 g/dL according to WHO criteria. Individual level variables were: sex, age, birth order, birth weight, religion, number of under-five children, childhood wasting, underweight, stunting, symptoms of acute respiratory infection, child fever and diarrhea, maternal anemia and age, parents’ educational and employment status, wealth index, source of drinking water, and type of toilet facility, whereas community-level variables were: Region, community-poverty, community-women education and community- women unemployment”

4.14. “We defined anemia in 6-59 month children as <11 g/dL according to WHO criteria (reference).” Please see earlier comments. Remember that unless you specify, I don’t even know that you are discussing children 6-59 months only.

We corrected it as per your comment as” We defined anemia in 6-59 month age children as hemoglobin <11 g/dL according to WHO criteria”.

4.15. Definitions are important. Consider using full sentences such as, “Wealth Index is a composite measure…” instead of : .

As per the comments you provided, we rewrite it as “Wealth index: is a composite measure of a household’s cumulative living standard. It was calculated based on household ownership of selected assets such as televisions and bicycles, cars; materials used for the housing construction; source of drinking water; and type of sanitation facilities. It was then generated using principal components analysis and the individual households were placed on a continuous scale of relative wealth. In the EDHS all mothers and children were assigned a standardized wealth index score. It was measured as a composite variable made up of five quintiles as poorest, poorer, middle, richer and richest.

4.16. Consider combining anthropometrics into a paragraph “Anthropometrics: Stunting was defined as height or length for age, HFA) <-2, wasting as ….

We corrected it as “Anthropometrics: Stunting was defined as height or length for age, (HFA) <-2SD (standard deviation), wasting as weight-for-height (WFH) <-2 SD and underweight as weight-for-age (WFA) <-2 SD.

4.17. What about children without a mother? Are they included? I wonder if some of the “mothers” are other relatives who are the child’s guardian.

Yes. We include children without a mother that is children with their guardian or caretakers. So the whole document is corrected by considering this comment.

4.18. What do you mean by “poor” or “poorest” etc.? Are these economic quintiles? Please define objectively this subjective term. This makes it difficult for me to interpret Figure 2.

Poor or poorest are economic quintiles. In the 2016 Ethiopia Demographic and Health Survey, these are quintiles of wealth index (economic quintiles) calculated based on household ownership of selected assets such as televisions and bicycles, cars; materials used for the housing construction; source of drinking water; and type of sanitation facilities. It was then generated using principal components analysis and the individual households were placed on a continuous scale of relative wealth. In the EDHS all mothers (caretakers) and children were assigned a standardized wealth index score. It was measured as a composite variable made up of five quintiles as poorest, poorer, middle, richer and richest 

Methods of data analysis

4.19. I would cut this section down to two paragraphs. The explanation of your model also needs to be condensed to one or two paragraphs. You may consider sharing your variable selection as an Appendix or supplement.

We aimed to explain more about the models to our reader, since it is unusual model especially in our country. But we annexed Table 1 as “annex 1” at the end of the main document.

5. Results

5.1. How many children were included?

We included 7790 children 6 -59 months of age who have hemoglobin test result

5.2. Please first state how many children you included and basic demographics (age, gender). Then state prevalence of anemia. Then go on to other variables.

We accepted the comment and we rewrite it again as” Seven thousand seven hundred ninety (7790) children 6 -59 months of age were included in this study. Above half (52%) of the children were male, and 34.2%, 33.3% and 32.2% where in the age category of 6-23, 24-41 and 42-59 months of age, respectively with mean ± SD (standard deviation) of 32 +15 months. The prevalence of anemia was 57.6% with a median hemoglobin concentration of 10.7 (IQR: 9.6-11.6)”. 

5.3. The sentence tells me that children were living with their partners. Surely this is not what is meant. Consider using “caretakers” to clarify when you discuss characteristics of the caretaker or mother.

We accepted the comment, and we rearranged the sentence as” Almost all of the respondents mothers (caretakers) (95%) were living with their respective partners and most of them were Muslims (40%) followed by Orthodox Christians (34%).

5.4. I would be more interested in seeing anemia sorted by economic status than a pie chart only of economic status. I think seeing child anemia sorted by maternal anemia (what is the definition of this, by the way?) would also provide useful information, as maternal anemia is a known risk factor for childhood anemia especially for the youngest.

All the significant variables have sorted by anemia and this is indicated in table 4.

Maternal anemia is defined as hemoglobin less than or equal to 11 g/dL (hg<=11 g/dL) according to WHO criteria. In the EDHS, hemoglobin levels were adjusted for pregnancy because during pregnancy the increase in maternal blood volume and the iron needs of the fetus decreases the blood Hb level.

5.5. Table 2. I think this should be Table 1. I would not present the current Table 1. Good to have this table (2), but some of this information can be presented as supplemental data. Please indicate the units, for example child age (months). Please consider labeling this as something like “Table x. Demographic characteristics of included children age 6-59 months selected from the 2016 EDHS.”

We annexed table 1, and we rename Table 2 as Table 1. We accept all the other comments and we amend our document accordingly.

5.6. Why does birth order 4-5 matter more than >=6?

According to our result presented in Table 4, birth order >=6 matter more than birth order 4-5. The odds of anemia were 1.26 (AOR=1.26; 95% CI: 1.00, 1.61) times higher for children with birth order six and above than first-order children (Table 4).

5.7. Community Level characteristics table: Please note this is mislabeled as Table 1. Please also see above suggestion for the title.

We corrected it accordingly.

Individual and community-level factors associated with anemia

5.8. Consider presenting the odds ratios in Table 1 next to the demographics. This would condense a lot of information into less space. Alternatively, cut down your demographics to the essential for Table 1 (currently labeled 2) and then keep Table 4 as is. Then present some of your data in graphs as well.

Table 1 (currently labeled 2) has presented simple frequency of all included variables. In addition, this table did not show any distribution of anemia by these factors, whereas Table 4 in the second column to right shows the anemia distribution by the significant factors. Then next to this, the Adjusted OR is written with their p-values. Therefore, these two tables are independent. 

5.9. Consider instead of simply stating all the OR, giving some context of importance. For example, “the strongest odds of anemia occurred with x and y (data).” Or “anemia was most strongly associated with ….”

We accepted the comment. we rewrite it as “From the individual-level factors, anemia was most strongly associated with child age, wealth index, maternal anemia and child stunting, whereas from the community-level, the strongest odds of anemia occurred among children from Somali, Harari, Dire Dawa and Afar region (Table 4).

5.10. Table 4: I would remove crude OR and present the adjusted OR with p value in the next column. Also, see above to cut down on demographics table as much of the same information is repeated here. Please explain what you adjusted for. Tell me more about the results of your multivariate model? Which variables made it into your model? This is not clear.

We removed crude OR and we present the adjusted OR with p value in the next column.

The demographic information written in Table 4 is not the same as the demographic information written in Table 1. Table 1 (previously labeled as Table 2) has presented simple frequency of all included variables. In addition, this table did not show any distribution of anemia by these factors, whereas Table 4 in the second column to right shows the anemia distribution by the significant factors. Then next to this, the Adjusted OR is written with their p-values.

5.11. Please explain what you adjusted for. Tell me more about the results of your multivariate model?

First bivariate analysis was performed to see the effect of each predictor variable on the outcome variable using a significance level of p<0.25 independently. Accordingly, anemia among children aged 6-59 months was associated with number of <5 children in the household, parents educational level and employment status, child age, religion of mother, birth order, maternal age, type of toilet facility , source of drinking water, wealth index, child stunting, wasting and underweight, fever, diarrhea, child deworming, symptoms acute respiratory infection, maternal anemia status, women community-education , place of residence, community-poverty , region and community- women unemployment status. Then all this variables were included simultaneously in to the final model to estimate adjusted odds ratios with 95% Confidence Interval (CI) at a significance level of p<0.05. 

In short, in the multivariable analysis the following variables were adjusted and controlled.

1. number of <5 children in the household 

2. child age

3. religion of mother

4. birth order

5. maternal and husband employment status

6. maternal age, type of toilet facility 

7. source of drinking water

8. wealth index

9. child stunting, wasting, underweight, fever, diarrhea 

10. child deworming, acute respiratory infection

11. maternal anemia status

12. community- women education 

13. place of residence, community-poverty 

14. region 

15. community- women unemployment 

After adjusting for all this variables, we found that anemia was most strongly associated with child age, wealth index, maternal anemia, child stunting, underweight, child fever, birth order, community-poverty and region (Somali, Harari, Dire Dawa, Afar, Oromia, Addis Ababa, Amhara and Benishangul). The result in 

Table 4 shows only the significant variables found at the multivariable analysis.

6. Discussion

6.1. “This study aimed to ….” (not was aimed to)

We correct it as” This study aimed to identify individual and community-level factors associated with anemia among children aged 6-59 months”.

6.2.“This finding indicated that” is too specific to one finding when you are discussing an entire study. Consider wording like: “We found that anemia was most strongly associated with … “ or “We found that … were the strongest predictors of anemia followed by…”

We accepted the comment and we rearranged as” We found that anemia among children aged 6-59 was most strongly associated with individual-level factors such as child age, wealth index, maternal anemia and child stunting followed by child underweight, child fever and birth order, whereas from the community-level the strongest odds of anemia occurred among children from Somali, Harari, Dire Dawa and Afar region followed by Oromia and Addis Ababa”.

6.3. Consider making a figure using a flowchart of the strongest predictors of anemia (from your model) with the adjusted OR or other information along the arrows that lead to anemia. This would be an excellent visual representation of your most important conclusion.

We corrected based on the, comment that order the strongest predictor in text, you provided to us. But the comments consider making a figure using a flowchart of the strongest predictors of anemia is not clear, sorry. 

6.4. What about confounders or collinearity? How confident are you that those factors did not influence your model?

We have checked confounders or collinearity to our model and no confounders or collinearity was detected. 

7. Conclusion

7.1. I would emphasize here the strongest of all predictors first.

We found that anemia among children aged 6-59 was most strongly associated with individual-level factors such as child age, wealth index, maternal anemia and child stunting followed by child underweight, child fever and birth order, whereas from the community-level the strongest odds of anemia occurred among children from Somali, Harari, Dire Dawa and Afar region followed by Oromia and Addis Ababa.

7.2. I think going from economic quartile as a risk factor to suggesting food subsidies is a big stretch. Perhaps say something like “our model predicts that … are most strongly associated with anemia in Ethiopian children 6-59 months with additional contribution from …. This suggests that interventions targeted to the improvement of …. may contribute to a reduction in childhood anemia and its devastating complications.”

Our model predicts that child age, maternal anemia and child stunting followed by child underweight, child fever and birth order and region are most strongly associated with anemia in Ethiopian children 6-59 months with additional contribution from wealth index. This suggests that interventions targeted to the improvement of Economic subsidy may contribute to a reduction in childhood anemia and its devastating complications.

Reviewer #2

Dear reviewer, thank you for your thorough review of our manuscript and for encouraging us.

---

## [Decision Letter · Decision Letter 1]

7 Oct 2020

PONE-D-20-09803R1

INDIVIDUAL AND COMMUNITY LEVEL FACTORS ASSOCIATED WITH ANEMIA AMONG CHILDREN 6 - 59 MONTHS OF AGE IN ETHIOPIA: A FURTHER ANALYSIS OF 2016 ETHIOPIA DEMOGRAPHIC AND HEALTH SURVEY

PLOS ONE

Dear Dr. Gebremeskel,

Thank you for submitting your manuscript to PLOS ONE. After careful consideration, we feel that it has merit but does not fully meet PLOS ONE’s publication criteria as it currently stands. Therefore, we invite you to submit a revised version of the manuscript that addresses the points raised during the review process.

Please consider the comments of reviewer 3 carefully, especially regarding the statistical analysis and subsequent interpretation.

We look forward to receiving your revised manuscript.

Kind regards,

Astrid M. Kamperman

Academic Editor

PLOS ONE

Reviewers' comments:

Reviewer's Responses to Questions

**Comments to the Author**

1. If the authors have adequately addressed your comments raised in a previous round of review and you feel that this manuscript is now acceptable for publication, you may indicate that here to bypass the “Comments to the Author” section, enter your conflict of interest statement in the “Confidential to Editor” section, and submit your "Accept" recommendation.

Reviewer #1: (No Response)

Reviewer #3: (No Response)

2. Is the manuscript technically sound, and do the data support the conclusions?

Reviewer #1: Yes

Reviewer #3: No

3. Has the statistical analysis been performed appropriately and rigorously? 

Reviewer #1: Yes

Reviewer #3: No

4. Have the authors made all data underlying the findings in their manuscript fully available?

Reviewer #1: Yes

Reviewer #3: Yes

5. Is the manuscript presented in an intelligible fashion and written in standard English?

Reviewer #1: Yes

Reviewer #3: Yes

6. Review Comments to the Author

Reviewer #1: Overall, much improved flow and language. Excellent abstract.

Statistics are still overly described, but that is your choice if you have the room for it.

Throughout:

The term “lower poverty status” is confusing. Are these communities less poor? I would say “less poverty” or “more poverty” instead of “higher poverty status” and vv. Another option is “lower wealth index”.

Check on small details like punctuation, missing brackets in tables around CI, spelling.

Re the use of “caretakers,” if mothers are mothers and there were no non-mother caretakers (ex. grandmothers), then there is no need to continue to say both words: mothers (caretakers).

I would use paragraph form and not bullets or check marks for inclusion/exclusion criteria, but that is an editorial choice.

Figure 2: the word “Characteristics” is misspelled

Tables:

I am of the opinion that tables should stand alone. If I look at these tables now without the rest of the article, I am lost and need more definitions and clarity.

For each table, give definitions in the captions. What do you mean by “low” education for example? How is frequency weighted? Give a description as a footnote/caption to the table.

What do you mean by community-women education or community-women unemployment? Are these grouped per region? If it is instead a community-level characteristic, then you have already labelled that in the title.

Discussion

First sentence of first paragraph has . and , in the middle of the sentence.

Great work.

Reviewer #3: The paper and analysis initially struck me as to offer interesting insight on individual and group level effects. On second look there are some omissions (e.g. in reporting on missing data) and some irregularities with what was reported about model building (checking for interaction effects). By looking at the data myself, I quickly find a mismatch between what the authors reported in their model building approach and what I find. I attach the R script I used and the descriptive plot I produced.

Aside from that, some revisions are required in formatting and presenting the results. Additionally I have left comments on some relatively small clarifications need to be added addressing choices the authors made when constructing and reporting on their model.

Please check the attached PDF file to access my notes on the manuscript.

7. PLOS authors have the option to publish the peer review history of their article (what does this mean?). If published, this will include your full peer review and any attached files.

Reviewer #1: No

Reviewer #3: **Yes: **Milan Zarchev

---

## [Author Response · Author response to Decision Letter 1]

13 Oct 2020

Responses to reviewers 

Reviewer #1

Dear reviewer, thank you for your thorough review of our manuscript. Your comments are constructive and very good lessons for us. We have tried to put our responses to your constructive comments and questions. The responses are put immediately after the questions, suggestions or comments. 

Reviewer #1: Overall, much improved flow and language. Excellent abstract.

Statistics are still overly described, but that is your choice if you have the room for it.

Throughout:

The term “lower poverty status” is confusing. Are these communities less poor? I would say “less poverty” or “more poverty” instead of “higher poverty status” and vv. Another option is “lower wealth index”.

We correct it based on your comment in the manuscript.

Check on small details like punctuation, missing brackets in tables around CI, spelling.

Re the use of “caretakers,” if mothers are mothers and there were no non-mother caretakers (ex. grandmothers), then there is no need to continue to say both words: mothers (caretakers).

We correct it based on your comment in the manuscript. 

I would use paragraph form and not bullets or check marks for inclusion/exclusion criteria, but that is an editorial choice.

We write it in paragraph form as” The inclusion criteria were children 6-59 months of age who live in the selected enumeration areas (community). And Exclusion criteria were children 6-59 months of age who have no hemoglobin test result.”

Figure 2: the word “Characteristics” is misspelled

We correct it.

Tables:

I am of the opinion that tables should stand alone. If I look at these tables now without the rest of the article, I am lost and need more definitions and clarity.

For each table, give definitions in the captions. What do you mean by “low” education for example? How is frequency weighted? Give a description as a footnote/caption to the table.

What do you mean by community-women education or community-women unemployment? Are these grouped per region? If it is instead a community-level characteristic, then you have already labelled that in the title.

We add footnote in to our manuscript based on your comment 

Discussion

First sentence of first paragraph has . and , in the middle of the sentence.

We correct it.

Great work.

Reviewer #3

Dear reviewer, thank you for your thorough review of our manuscript and for encouraging us. 

Reviewer #3: The paper and analysis initially struck me as to offer interesting insight on individual and group level effects. On second look there are some omissions (e.g. in reporting on missing data) and some irregularities with what was reported about model building (checking for interaction effects). By looking at the data myself, I quickly find a mismatch between what the authors reported in their model building approach and what I find. I attach the R script I used and the descriptive plot I produced.

The R script cannot open. So we cannot see it, sorry.

1. Reporting on missing data

From the total of 10,641 under-five years’ old children, 9504 were children 6-59 months of age. Data on hemoglobin test result from the survey were available for 7790 children. As a result, 1714 children aged 6-59 month were excluded from the study due to missing data of hemoglobin test result. In addition, the variables of dietary intake and child feeding practices were not included due to missing value. These variables were missing for nearly half of observations. This is due to the reason that our study sample was age 6-59 months that the variables of dietary intake and child feeding practices were not a concern for children less than 6 months old. Other possible child-related explanatory variables such as parasitic infection and chronic illness were not included in the analysis because these variables are not in the EDHS data. 

2. Regarding Interaction

Interaction between variables was checked for those variables found significant at the final model. As a result, there were significant interactions (p<0.05) between these variable. However, as we examined the interaction effect by fitting regression models that contained interaction terms yields no significant (p>0.05) interaction effect. “We have attached evidences as supplementary material”

Aside from that, some revisions are required in formatting and presenting the results. Additionally I have left comments on some relatively small clarifications need to be added addressing choices the authors made when constructing and reporting on their model.

We corrected it based on the comments you provided on the manuscript.

---

## [Decision Letter · Decision Letter 2]

20 Oct 2020

INDIVIDUAL AND COMMUNITY LEVEL FACTORS ASSOCIATED WITH ANEMIA AMONG CHILDREN 6 - 59 MONTHS OF AGE IN ETHIOPIA: A FURTHER ANALYSIS OF 2016 ETHIOPIA DEMOGRAPHIC AND HEALTH SURVEY

PONE-D-20-09803R2

Dear Dr. Gebremeskel,

We’re pleased to inform you that your manuscript has been judged scientifically suitable for publication and will be formally accepted for publication once it meets all outstanding technical requirements.

Kind regards,

Astrid M. Kamperman

Academic Editor

PLOS ONE

Additional Editor Comments (optional):

Reviewers' comments:

Reviewer's Responses to Questions

**Comments to the Author**

1. If the authors have adequately addressed your comments raised in a previous round of review and you feel that this manuscript is now acceptable for publication, you may indicate that here to bypass the “Comments to the Author” section, enter your conflict of interest statement in the “Confidential to Editor” section, and submit your "Accept" recommendation.

Reviewer #3: All comments have been addressed

2. Is the manuscript technically sound, and do the data support the conclusions?

Reviewer #3: Yes

3. Has the statistical analysis been performed appropriately and rigorously? 

Reviewer #3: Yes

4. Have the authors made all data underlying the findings in their manuscript fully available?

Reviewer #3: Yes

5. Is the manuscript presented in an intelligible fashion and written in standard English?

Reviewer #3: Yes

6. Review Comments to the Author

Reviewer #3: All comments were either incorporated or addressed adequately. I have no further major recommendations for this paper.

7. PLOS authors have the option to publish the peer review history of their article (what does this mean?). If published, this will include your full peer review and any attached files.

Reviewer #3: **Yes: **Milan Zarchev

---

## [Editor Report · Acceptance letter]

4 Nov 2020

PONE-D-20-09803R2 

INDIVIDUAL AND COMMUNITY LEVEL FACTORS ASSOCIATED WITH ANEMIA AMONG CHILDREN 6 - 59 MONTHS OF AGE IN ETHIOPIA: A FURTHER ANALYSIS OF 2016 ETHIOPIA DEMOGRAPHIC AND HEALTH SURVEY 

Dear Dr. Gebremeskel:

I'm pleased to inform you that your manuscript has been deemed suitable for publication in PLOS ONE. Congratulations! Your manuscript is now with our production department. 

Kind regards, 

on behalf of

Dr. Astrid M. Kamperman 

Academic Editor

PLOS ONE